# Research on germplasm diversity of *Amomum villosum*. Lour in genuine producing area

Jie Xu[1,2◉], Bohan Yang[1◉], Mingxiao Li[1,3], Zixiang Li[1], Yuting Tu[1], Liyun Tang[1,4], Guozhen He[1]*

1 School of Traditional Chinese Medicine, Guangzhou University of Chinese Medicine, Guangzhou, Guangdong, China, 2 Guangdong Yifang Pharmaceutical Co., Ltd., Guangdong Provincial Key Laboratory of Traditional Chinese Medicine Formula Granule, Foshan, Guangdong, China, 3 Zhuhai Hospital of Integrated Traditional Chinese and Western Medicine, Zhuhai, Guangdong, China, 4 School of Life Science, South China Agricultural University, Guangzhou, Guangdong, China

◉ These authors contributed equally to this work.
* heguozhen@gzucm.edu.cn

## Abstract

### Background

Genuine Chinese medicine is produced from medicinal plant cultivated in a specific region and is of better quality and efficacy, more consistently qualified and famous than that from the same medicinal plant cultivated in other regions. The cultivating region of genuine medicinal plant is known as the genuine producing area. Yangchun City, which is in Guangdong Province of China, is a genuine producing area for the famous Chinese medicine Amomi Fructus (also called Sharen). Amomi Fructus is the ripe and dry fruit of the Zingiberaceae plant *A. villosum* Lour.. *A. villosum* was introduced from the Persian Gulf region and has been cultivated in China for over 1000 years. Until now there are no reports on screening for good germplasm of *A. villosum*.

### Methods

The contents of volatile oil and bornyl acetate of Amomi Fructus from 14 populations were determined with GC method, and the relative contents of the main chemical components in the volatile oils were determined with GC-MS method. Evaluation and variance analysis of the comprehensive quality of the 14 samples were conducted by means of a multi-indicator entropy-weight TOPSIS model (Technique for Order Preference by Similarity to an Ideal Solution) combined with OPLS-DA (Orthogonal Partial Least Squares Discrimination Analysis) and HCA (Hierarchical Clustering Analysis). The ISSR (Inter-Simple Sequence Repeat) molecular marker technique and the UPGMA (unweighted pair-group method with arithmetic means) were employed to analyze the genetic relationship among *A. villosum* populations.

### Results

The contents of volatile oil and bornyl acetate differed significantly among the different populations, but the main chemical component in the volatile oil was the same in all the samples,

**Data Availability Statement:** All relevant data are within the paper and its Supporting Information files.

**Funding:** This research was supported supported by the Key-Area Research and Development

Program of Guangdong Province (2020B020221002), the Forestry Science and Technology Innovation Project of Guangdong, China (2018KJCX033), the Agricultural Science and Technology System Management Project of Guangdong, China (0835-190Z22404211) and the Innovation team building Project of South medicinal industry in Modern Agricultural and Industrial Technology System, Guangdong China (2020KJ142, 2019KJ142) "The funder provided support in the form of salaries for author [JX], but did not have any additional role in the study design, data collection and analysis, decision to publish, or preparation of the manuscript. The specific roles of these authors are articulated in the 'author contributions' section."

**Competing interests:** The authors have declared that no competing interests exist.

which was bornyl acetate. OPLS-DA results showed that 9 indicators were the main factors influencing the quality differences among the 14 populations. The entropy-weight TOPSIS results showed that there were significant differences in the comprehensive qualities of the 12 populations from the genuine producing area. The best quality of fruit was found in the genuine producing area of Chunwan Town; the qualities of 33% of genuine fruits were lower than that of non-genuine fruits. Twenty-three DNA fragments were obtained by ISSR-PCR amplification using four ISSR primers, eleven of which were polymorphic loci, which accounted for 47.8%. The similarity coefficients (GS) of different populations of *A. villosum* ranged from 0.6087 to 0.9565.

## Conclusion

There are significant differences among different populations of *A. villosum* in terms of the kinds of major chemical components and their contents, comprehensive quality and genetic diversity. The germplasm resources of *A. villosum* are rich in the genuine producing area. It means superior germplasm could be selected in the area. The comprehensive quality of the fruit of *A. villosum* from the non-genuine producing area is better than some of that from genuine producing area, proving that the non-genuine producing area can also produce Amomi Fructus with excellent quality.

## Introduction

*Amomum villosum*. Lour is a perennial evergreen herb belonging to *Amomum* in Zingiberaceae family, which is mainly distributed in the provinces of Guangdong, Yunnan and Hainan in China and other places [1]. The *A. villosum* in Guangdong Province was introduced from the Persian Gulf region and has been cultivated for over 1,000 years; the *A. villosum* in the other four places was introduced from Guangdong Province, with the longest cultivation history not exceeding 60 years [2].

The dried capsule of *A. villosum* is brown, ellipsoid, with soft spines on the surface, which has the effect of dampening appetite, "warming" the spleen to stop diarrhea, regulating Qi, and preventing miscarriage. It is one of the famous "four southern medicines" in China [3, 4]. Studies have shown that the volatile oil in the fruits of *A. villosum* is the main component of its medicinal effect. Twenty-one chemical components have been identified in the volatile oil of the fruits of *A. villosum*, including high levels of bornyl acetate, camphor and borneol [5]. Among them, bornyl acetate not only accounts for the largest proportion of the volatile oil, but also is the main active ingredient, with anti-inflammatory and analgesic [6–8], antioxidant [9] and immunomodulatory pharmacological effects [10]. Therefore, bornyl acetate is considered to be the main active component of the volatile oil in *A. villosum*.

Yangchun City of Guangdong Province is considered as the genuine producing area of *A. villosum* due to the high quality of Amomi Fructus and long cultivating history of *A. villosum* [1]. *A. villosum* is a sexually reproductive plant and there is a possibility that the genetic traits of *A. villosum* have been changed because of long-term cultivating in the genuine producing area. We have found four cultivated types of *A. villosum* in the genuine producing area: *A. villosum* cv. Changguo, *A. villosum* cv. Yuanguo, *A. villosum* cv. Zhonghua and *A. villosum* cv. Jinqiu [11], which confirms that the existence of germplasm diversity is in the genuine producing area. However, there is a lack of detailed studies on the richness of germplasm diversity

and the difference in quality and germplasm between genuine *A. villosum* and non-genuine *A. villosum*.

Technique for order preference by similarity to ideal solution (TOPSIS) is a multi-objective decision-making analysis method that calculates the closeness of multiple indicators to the optimal solution and prioritizes the objects to be evaluated. It can eliminate the interference of subjective weighting and multi-type variables on decision-making, and the results of comprehensive evaluation are more scientific and accurate [12]. The entropy-weighted TOPSIS has been widely used in the comprehensive multi-indicator evaluation of *Bupleurum chinense* DC. [13], *Panax notoginseng* [14], *Cynomorium songaricum* [15] and other Chinese herbs, which can provide guidance for the selection of Chinese herbs for cultivation.

In recent years, DNA-based molecular markers have become necessary to study the germplasm diversity of certain traditional Chinese medicine, and various molecular markers such as random amplified polymorphic DNA (RAPD) [16, 17], simple sequence repeats (SSR) [18, 19] and inter-simple sequence repeat (ISSR) [20–22] have been used to distinguish between different germplasm of the same herb. Compared with other technologies, ISSR is simple to operate, rich in polymorphism, reproducible and stable [23, 24], and has been widely used in plant variety identification, genetic diversity analysis, molecular breeding and other fields [25, 26] In this paper, the genetic diversity of *A. villosum* from different populations of genuine producing area and non-genuine producing area was analyzed using ISSR techniques, and the comprehensive quality of the collected samples was assessed by combining with TOPSIS model, with the aim of providing a theoretical basis for screening out excellent germplasm of *A. villosum* and expanding the cultivation range of excellent germplasm for the healthy development of *A. villosum* industry.

## Material and methods

### Plant material and chemicals

In this paper, we investigated the germplasm diversity of *A. villosum* in order to screen for good germplasm in the future. In this study, we selected 12 populations of *A. villosum* in the genuine producing region and two other populations of *A. villosum* in the non-genuine producing area of Shitang Town, Renhua County, Shaoguan City, Guangdong Province, and collected their fruits and fresh leaves for quality evaluation and genetic variation detection. The sampled plants were identified by Guozhen He, Professor of College of Traditional Chinese Medicine, Guangzhou University of Traditional Chinese Medicine. Fresh healthy leaves and mature fruits were taken from different plants within each population, and the leaves were stored in liquid nitrogen immediately after collection in the field and then stored in the ultralow temperature refrigerator in the laboratory. Details of the samples collected are given in Table 1. Of these, S13 and S14 were introduced to Shitang Township in 2010 from Chunwan Township, Yangchun City, in the village of Oudong. Bornyl acetate (BA, purity: >98%), the main constituent in volatile oil of *A. villosum* Lour, was purchased from National Institutes for Food and Drug Control of China. Pictures of the fruit of each population are shown in Fig 1.

### Preparation and detection of Volatile Oil of *A. villosum* (VOAV) and Bornyl Acetate of *A. villosum*(BAAV)

VOAV was extracted from the powdered crude drug by steam distillation according to the procedure recorded in the Chinese Pharmacopoeia (2020 edition) [4]. The powdered (15 g, 20 mesh) seed mass of *A. villosum* was weighed, and then steam distillation was performed with 500 mL of water for 5 h, left for 1 hour to calculate the content of volatile oil.

**Table 1. Collection information of 14 different populations of *A. villosum*.**

| No. | Pop. | Longitude(N) | Latitude(E) | Sample Size | Origin |
|---|---|---|---|---|---|
| 1 | S1 | 112°0' 33" | 22°17' 23" | 23 | Heshui Town, Yangchun City, Guangdong Province (HS) |
| | S2 | 112°1' 18" | 22°17' 08" | 25 | |
| | S3 | 112°1' 21" | 22°16 54" | 27 | |
| | S7 | 112°01′24″ | 22°17′06″ | 26 | |
| | S8 | 112°01′20″ | 22°16′59″ | 25 | |
| | S9 | 112°01′15″ | 22°17′04″ | 25 | |
| | S10 | 112°01′05″ | 22°17′13″ | 22 | |
| | S11 | 112°01′03″ | 22°17′03″ | 24 | |
| | S12 | 112°01′13″ | 22°17′21″ | 25 | |
| 2 | S4 | 112°3' 15" | 22°23' 53" | 25 | Chunwan Town, Yangchun City, Guangdong Province (CW) |
| | S5 | 112°3' 23" | 22°23' 54" | 26 | |
| 3 | S6 | 111°55' 48" | 22°11' 05" | 25 | Panlong village, Yangchun City, Guangdong Province (PL) |
| 4 | S13 | 112°0' 33" | 22°17' 23" | 28 | Shitang Town, Shaoguan City, Guangdong Province (SG) |
| | S14 | 112°1' 18" | 22°17' 08" | 28 | |

Powdered (1 g, 50 mesh) seed mass of *A. villosum* was weighed, added 25 mL of absolute ethanol, sonicated (300 W, 40 kHz) for 30 min, cooled at room temperature and absolute ethanol was used to make up the lost weight, filtered and the filtrate was taken after filtration, which was the test substance solution; and then an appropriate amount of bornyl acetate control sample was taken to prepare a solution of 0.3 mg/mL, which was the control sample

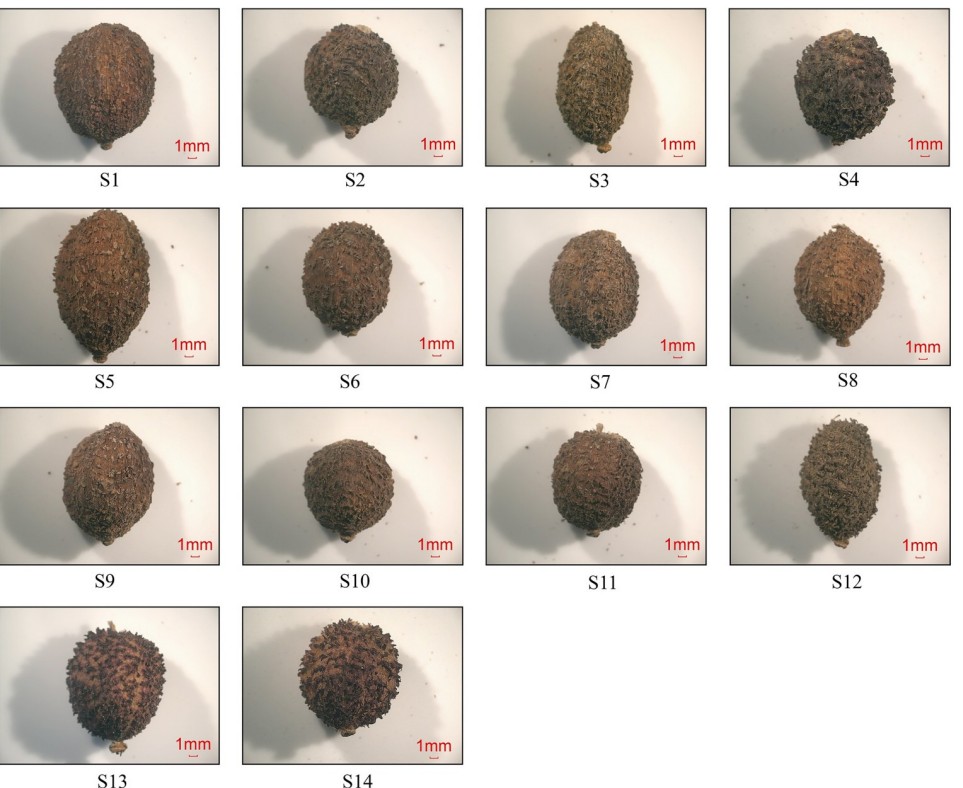

**Fig 1. Illustration of the fruit of *A. villosum* from different populations.**

solution, and 1 μL each was taken for determination. All experiments were conducted using Agilent Technologies 7890B system(Agilent Technologies, Santa Clara, California, USA). These compounds were separated by a HP-5MS Capillary column(30 m × 0.25 mm × 0.25 μm, Agilent Technologies, USA), the split ratio was 10:1, the inlet temperature, detector (FID) temperature and column temperature were set at 230˚C, 250˚C, 100˚C, respectively.

## Quality determination of VOAV by Gas Chromatography-Mass Spectrometer (GC-MS)

GC-MS (Agilent Technologies, Santa Clara, CA, USA) was used for chromatographic analysis to determine the quality of VOAV with a capillary column (30 m × 0.25 mm × 0.25 μm). The oven temperature was programmed as follows: an initial temperature of 50˚C, which was increased to 150˚C at a rate of 4˚C min$^{-1}$, kept 2 min, and then 8˚C min$^{-1}$ to final temperature of 250˚C. Injection was conducted in split mode (20:1) at 230˚C. The carrier gas helium was at a flow rate of 1.0 mL/min, and the injected sample volume was 1 μL. Under electron impact (EI) ionization (70 EV), the MS scanning range is (M/z) 60–600 atomic mass units (AMU). The EI source and quadrupole temperatures were set to 230˚C and 150˚C, respectively. The results obtained were compared and analyzed with the library (NIST08), and the matching degree > 90 was used as the screening basis for compound identification and the area normalization method was used to measure the relative mass percentage of each component.

## Determination of biological characters of fruit and seed mass of *A. villosum*

The collected fresh fruits of *A. villosum* were placed in blast drying oven at 50˚C to dry. The fresh weight, dry weight, seed mass, seed mass dry weight, thousand-grain fresh weight (seed), and thousand-grain dry weight (seed) of each population of *A. villosum* fruits were measured by electronic balance (Sartorius, Göttingen, Germany), and the vernier calipers were used to measure the horizontal and vertical diameters of dried fruits and other traits. The data obtained were statistically analyzed by SPSS 23.0 software and One-way ANOVA analysis of variance, expressed as means ± SE.

## Analysis on the difference of *A. villosum* quality among different populations

Orthogonal partial least squares discriminant analysis (OPLS-DA) is a multivariate statistical data analysis method, whose most important feature is that the categorical information can be concentrated mainly on one principal component, the model becomes simple and easy to solve, and its discriminant effect and visualisation of the principal component score plot are more obvious [27, 28]. In SIMCA14.1 software, the quality differences of 14 *A. villosum* samples were analyzed with the methods of OPLS-DA and HCA with the contents of *A. villosum* volatile oil and bornyl acetate, the relative contents of the main components in volatile oil, and the biological traits of fruit and seed mass as variables. The variable importance for the projection (VIP) for the fruit and seed mass of different populations of *A. villosum* was ranked and those with VIP values greater than 1 were selected as the main indicators for differentiating the quality differences among different populations of *A. villosum*.

## Comprehensive quality evaluation of *A. villosum* in various populations

**Establishment of standardized decision matrix.** Due to the different dimensions of different indexes of *A. villosum*, it is necessary to homogenize the data, establish a standardized decision matrix, and normalize the standardized decision matrix at the same time. Because the

evaluation indicators involved in this study were all high-quality indicators, the obtained data was processed with high-quality indicators in the same direction according to the formula (1), $X_{ij}$ is the measured value of each sample $i$ under each index $j$.

$$R_{ij} = \left[ \frac{X_{ij} - \min(X_{1j}, X_{2j} \ldots, X_{mj})}{\max(X_{1j}, X_{2j} \ldots, X_{mj}) - \min(X_{1j}, X_{2j} \ldots, X_{mj})} \right] \tag{1}$$

**Weight determination.** Entropy method weighting can eliminate subjective influence and objectively reflect the importance of evaluation indicators. The calculation of entropy and weight follows formula (2), $E_j$ refers to the entropy of the $j$-th index, and $w_j$ refers to the weight of the $j$-th index.

$$E_j = -k \sum_{i=1}^{m} R_{ij} \ln R_{ij}$$

$$k = \frac{1}{\ln m}, \text{ When } R_{ij} = 0, \text{ let } R_{ij} \ln R_{ij} = 0$$

$$w_j = \frac{1 - E_j}{\sum_{j=1}^{n}(1 - E_j)} \left( \sum_{j=1}^{n} w_j = 1 \right) \tag{2}$$

**Construction of weighted decision matrix.** The calculation of the weighted decision matrix follows formula (3), and the calculation of the optimal vector $Z^+$ and the worst vector $Z^-$ is shown in formulas (4) and (5).

$$Z = (R_{ij} \times w_j)_{m \times n} \tag{3}$$

$$Z_i^+ = \max(Z_{1j}, Z_{2j}, \ldots, Z_{nj}) \tag{4}$$

$$Z_j^- = \min(Z_{1j}, Z_{2j}, \ldots, Z_{nj}) \tag{5}$$

**Nearness degree calculation and evaluation.** The distance $D$ between the sample to be evaluated and the optimal vector $Z^+$ and the worst vector $Z^-$ and the Euclid approach degree $C_i$ with the best solution is calculated according to the formula (6) and formula (7), and then it is ranked. If $C_i$ is within [0,1], the larger the evaluation object is, the closer it is to the optimal level, and the smaller the evaluation object is, the closer it is to the worst level.

$$D_i^+ = \sqrt{\sum_{j=1}^{n} (z_{ij} - z_j^+)^2}, \ D_i^- = \sqrt{\sum_{j=1}^{n} (z_{ij} - z_j^-)^2} \tag{6}$$

$$C_i = \frac{D_i^-}{D_i^+ + D_i^-} \tag{7}$$

## Analysis of the genetic relationship between different populations of *A. villosum* based on ISSR

**DNA extraction and ISSR-PCR amplification system.** The extraction of DNA from the leaves of each population of *A. villosum* was taken using the DP305 plant DNA kit (TIANGEN Biotech Co., Ltd., Beijing, China). The NanoDrop2000 ultra-micro ultraviolet

spectrophotometer (Thermo Scientific, MIT, USA) was used to determine the DNA concentration and purity. Ten selected ISSR primers were used in the germplasm identification study of *A. villosum* [29, 30], and from these, four primers that produce clear and repeatable band patterns were selected, 14 populations of *A. villosum* samples were amplified by ISSR-PCR. The 20μL ISSR reaction volume included 2 μL Ex Taq Buffer, 2 μL template DNA (20 ng), 15 mmol·L$^{-1}$ MgCl2, 25 ng DNA template, 0.5 μmol·L$^{-1}$ ISSR primer, 0.75 mmol·L$^{-1}$ dNTPs, 1.0 U Ex Taq, sterile water to make up the volume. The ISSR PCR amplification was programmed in the PTC-200 thermocycler (Bio-Rad, Hercules, CA, USA) as follows: pre-denaturation at 94˚C for 5 min, 35 cycles of denaturation at 94˚C for 30 s, annealing at the 50.0–53.0˚C according to the primers for 40 s, extension at 72˚C for 90 s, with a final extension at 72˚C for 7 min and preservation at 4˚C. After electrophoresis in 1 × TAE buffer at 120 V for 1.5 h, ISSR-PCR products were separated on a 2% agarose gel stained with Goldview, and then the gel was visualized under UV light and photographed with a Mshot MSX2 imaging system (MINGMEI Co., Ltd., Guangzhou, China).

**ISSR-based data analysis.** Clear bands were scored as either present (1) or absent (0) in electropherogram, thus generating an ISSR phenotype data matrix that was imported into NTSYS 2.10e software to analyze genetic identity, and clustering dendrogram was constructed by unweighted pair-group method with arithmetic mean (UPGMA) [31].

## Results

### Comparison of the contents of VOAV and BAAV in various populations

Here, we extracted VOAV and BAAV from seed mass of *A. villosum* from 14 populations (Table 2). Except for population S14, the VOAV content of all populations could reach the standard of Chinese Pharmacopoeia (not less than 3.0%); the content of BAAV in each population was much higher than the standard of Chinese Pharmacopoeia (not less than 0.9%). However, there was a great difference among the populations, and the content of bornyl acetate in population S12 and population S2 was 1.9 times different, which indicated that there was a big difference in the content of active ingredients in the seed mass of *A. villosum* in different populations in Genuine producing area.

### Analysis of chemical composition in VOAV

Six major characteristic peaks were identified by GC-MS in the VOAV (Fig 2), and the relative contents were obtained by their peak area ratios, and these data are listed in Table 3. In Table 3, the components with higher contents in the volatile oil were found to be camphor, borneol and bornyl acetate, respectively. The relative content of bornyl acetate is the highest,

**Table 2. Content comparison of volatile oil or bornyl acetate among different populations of *A. villosum*(n = 3).**

| Population | Content of volatile oil(%) | Contents of bornyl acetates(%) | Population | Content of volatile oil (%) | Contents of bornyl acetates(%) |
|---|---|---|---|---|---|
| S1 | 3.88±0.21 a | 2.83±0.07 a | S8 | 3.03±0.09 bhi | 2.62±0.09 c |
| S2 | 3.00±0.05 bc | 1.60±0.22 b | S9 | 3.00±0.10 cgi | 1.57±0.12 b |
| S3 | 3.49±0.34 def | 2.61±0.12 c | S10 | 3.14±0.21b fgh | 2.90±0.05 ag |
| S4 | 3.36±0.26 bd | 1.95±0.05 de | S11 | 3.34±0.07 bdg | 2.35±0.08 h |
| S5 | 3.58±0.23 ad | 2.14±0.16 df | S12 | 3.80±0.26 ae | 3.04±0.15 g |
| S6 | 3.26±0.17 bdg | 2.06±0.03 def | S13 | 3.17±0.19 bfgh | 2.25±0.06 fh |
| S7 | 3.37±0.42 dh | 1.98±0.23 de | S14 | 2.68±0.05 ci | 1.90±0.03 e |

Note: No letters or different letters in the same column indicate significant difference ($P < 0.05$), the same below.

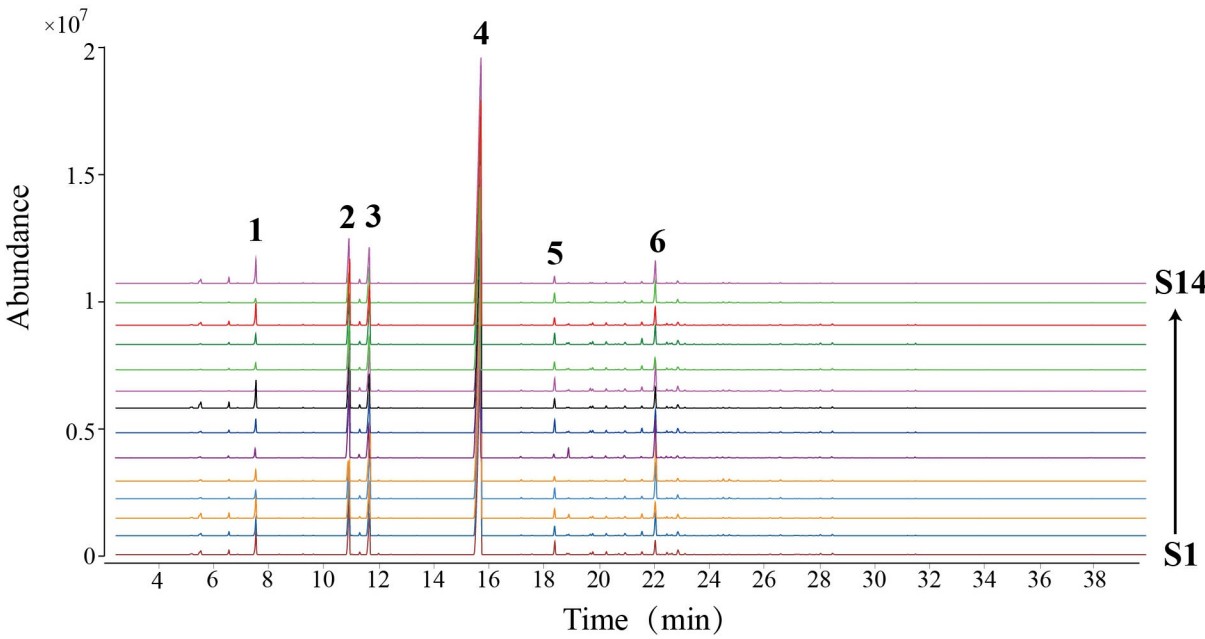

**Fig 2. GC-MS chromatogram of 14 populations of *A. villosum*.** 1. Limonene; 2. Camphor; 3. Borneol; 4. Bornyl acetate; 5. α- Copaene; 6. Compound 6.

ranging from 73.98% to 83.92%, with an average of 77.03±2.78%, followed by the content of camphor, with an average of 8.50±3.16%, while the relative content of bornyl acetate far exceeds the sum of the other components, and can be identified as the representative component of VOAV. S13 and S14 were cultivated in the same producing area (Shitang Town), but S13 contained limonene, while S14 did not, indicating that the two are different germplasms of *A. villosum*. The same result also occurred in populations S4 and S5, both from Chunwan Town, and the former contained α-copaene, while the latter did not.

## Difference of biological characters of the fruit and seed mass of *A. villosum*

There were great differences in the characteristics of *A. villosum* from different populations (Table 4). The fresh weight of the fruit of populations S1, S5 and S9 was significantly higher than that of the other populations. The difference between the highest fresh weight population S1 and the lowest fresh weight population S7 was 1.45 times higher, but the drying rate of fruits (dry weight / fresh weight) of S1 was only 27%, indicating its relatively high moisture content. Differences in seed mass indicators between populations were also evident. Although there was a 1.51-fold difference between S3, which had the lowest fresh weight, and S5, which had the highest fresh weight, S3 had the highest percentage of drying rate of fruits at 55%.

After comparing the morphology of the fruits of different populations of *A. villosum* after drying, it was found that the transverse longitudinal diameter ratio of the dried fruits of each population was between 0.69~0.88, Populations S5 had the smallest transverse longitudinal diameter ratio of fruit at 0.69, the fruits of S3 and S12 were differed significantly from the ratios of the other 12 populations. In terms of fruit shapes of *A. villosum*, S1, S2, S4, S6, S7, S8, S9, S10, S11, S13 and S14 were probably the same germplasm, S5 was a class of germplasm, and S3 and S12 were an in-between class of germplasm.

**Table 3. Comparison of characteristic components of volatile oil in different *A. villosum* populations.**

| Population No. | Testing index | Compound | | | | | |
|---|---|---|---|---|---|---|---|
| | | Limonene ($C_{10}H_{16}$) | Camphor ($C_{10}H_{16}O$) | Borneol ($C_{10}H_{18}O$) | Bornyl acetate ($C_{10}H_{12}O_2$) | α-Copaene ($C_{15}H_{14}$) | Compound 6 ($C_{15}H_{24}$) |
| S1 | A | 7.19 | 10.58 | 11.32 | 15.37 | 18.04 | 21.68 |
| | B | 5943034 | 16546797 | 13700731 | 147170347 | 3163950 | 3747056 |
| | C | 3.08 | 8.56 | 7.09 | 76.13 | 1.64 | 1.94 |
| S2 | A | 7.18 | 10.58 | 11.32 | 15.36 | 18.03 | 21.70 |
| | B | 4001596 | 18675465 | 13887770 | 144124512 | 2340752 | 6558752 |
| | C | 2.11 | 9.85 | 7.33 | 76.02 | 1.23 | 3.46 |
| S3 | A | 7.18 | 10.58 | 11.30 | 15.36 | 18.02 | 21.68 |
| | B | 5769138 | 19053069 | 11812962 | 149543185 | 2310737 | 4411954 |
| | C | 2.96 | 9.77 | 6.06 | 76.72 | 1.19 | 2.26 |
| S4 | A | 7.178 | 10.547 | 11.328 | 15.368 | 18.03 | 21.707 |
| | B | 1936307 | 8451857 | 1821763 | 157999383 | 2636536 | 10645811 |
| | C | 0.97 | 4.23 | 9.11 | 79.04 | 1.32 | 5.33 |
| S5 | A | 7.18 | 10.53 | 11.33 | 15.37 | - | 21.71 |
| | B | 2621148 | 4520151 | 21060571 | 157251232 | - | 11914682 |
| | C | 1.33 | 2.29 | 10.67 | 79.67 | - | 6.04 |
| S6 | A | 7.18 | 10.60 | 11.30 | 15.35 | 18.55 | 21.71 |
| | B | 2052438 | 24657036 | 9476438 | 134322079 | 2329662 | 11046629 |
| | C | 1.12 | 13.41 | 5.16 | 73.05 | 1.26 | 6.00 |
| S7 | A | 7.184 | 10.584 | 11.31 | 15.362 | 18.03 | 21.689 |
| | B | 3209934 | 21146396 | 12416254 | 150444397 | 3049982 | 6191255 |
| | C | 1.63 | 10.77 | 6.32 | 76.58 | 1.55 | 3.15 |
| S8 | A | 7.19 | 10.59 | 11.30 | 15.35 | 18.02 | 21.68 |
| | B | 6883613 | 22218258 | 10530080 | 138900430 | 2244109 | 5696684 |
| | C | 3.62 | 11.68 | 5.53 | 72.98 | 1.18 | 2.99 |
| S9 | A | 7.178 | 10.56 | 11.291 | 15.313 | 18.024 | 21.67 |
| | B | 1611896 | 12685017 | 9690648 | 97557235 | 1860737 | 3099488 |
| | C | 1.25 | 9.87 | 7.54 | 75.90 | 1.45 | 2.41 |
| S10 | A | 7.18 | 10.57 | 11.32 | 15.36 | 18.02 | 21.68 |
| | B | 2368467 | 15634404 | 16350636 | 141037692 | 2714538 | 5390162 |
| | C | 1.28 | 8.46 | 8.84 | 76.25 | 1.46 | 2.91 |
| S11 | A | - | 10.56 | 11.32 | 15.36 | 18.03 | 21.70 |
| | B | - | 12124386 | 15024182 | 147992687 | 3067522 | 8025227 |
| | C | - | 6.45 | 8.00 | 78.81 | 1.63 | 4.27 |
| S12 | A | 7.18 | 10.59 | 11.3 | 15.35 | 18.02 | 21.68 |
| | B | 5338709 | 23163840 | 13012972 | 141084241 | 1807460 | 4877978 |
| | C | 2.82 | 12.24 | 6.87 | 74.54 | 0.95 | 2.58 |
| S13 | A | 7.18 | 10.56 | 11.30 | 15.36 | 18.02 | 21.68 |
| | B | 5837615 | 13353366 | 11166219 | 148023665 | 1675289 | 6095485 |
| | C | 3.11 | 7.11 | 5.94 | 78.81 | 0.89 | 3.25 |
| S14 | A | - | 10.54 | 11.30 | 15.35 | 18.02 | 21.68 |
| | B | - | 7046535 | 11213235 | 138041282 | 2340197 | 5844406 |
| | C | - | 4.28 | 6.81 | 83.93 | 1.43 | 3.55 |

Note: A: Retention time (min), B: Peak area, C: Relative content (%). Compound 6 is (1S, 2E, 6E, 10R)-3, 7, 11, 11-tetramethylbicyclo [8.1.0] undec-2,6-diene, "-" indicates that the compound was not detected.

**Table 4. Comparison of characteristics among different populations of *A. villosum*.**

| Population | Fresh weight of 100 fruits(g) | Dry weight of 100 fruits(g) | Fresh weight of seed mass (g) | Dry weight of seed mass (g) | Fresh weight of 1000 seeds (g) | Dry weight of 1000 seeds(g) | Drying rate of fruits(%) | Transverse diameter(mm) | Longitudinal diameter(mm) | Transverse longitudinal diameter ratio |
|---|---|---|---|---|---|---|---|---|---|---|
| S1 | 275.83 ±12.2a | 75.29 ±0.45a | 1.29±0.06ab | 0.62±0.02a | 32.53 ±0.13ab | 12.73 ±0.12a | 27.33 ±1.11a | 12.41±0.14ab | 14.61±0.45ab | 0.85±0.03ab |
| S2 | 255.30 ±5.26bc | 80.06 ±1.35bf | 1.40±0.05ac | 0.72±0.02b | 27.54±1.16c | 13.36 ±0.17b | 31.36 ±0.17b | 12.39±0.05ab | 14.09±0.45ac | 0.88±0.03a |
| S3 | 211.01 ±6.42d | 66.04 ±0.10cd | 1.00±0.02d | 0.55±0.01cd | 37.65±0.87d | 15.05±0.05 | 31.32 ±0.98bc | 10.98±0.25c | 14.57±0.21ab | 0.75±0.02c |
| S4 | 245.92 ±9.20bef | 59.89 ±1.07e | 1.15±0.10e | 0.49±0.03c | 28.23 ±2.51ce | 13.04 ±0.15c | 24.38±1.15 | 12.40±0.17ab | 14.27±0.04abd | 0.87±0.01ad |
| S5 | 266.27 ±11.80ac | 78.37 ±0.71f | 1.51±0.06cf | 0.70±0.01be | 27.39±0.55c | 12.61 ±0.22ad | 29.48 ±1.55de | 11.55±0.16de | 16.68±0.37 | 0.69±0.02e |
| S6 | 213.82 ±5.42d | 67.45 ±1.64cg | 1.31±0.06ag | 0.60±0.03af | 20.19±0.57 | 9.20±0.25 | 31.55 ±0.67b | 11.66±0.19dfg | 14.61±0.27ab | 0.80±0.02f |
| S7 | 189.65 ±9.13g | 59.09 ±1.05e | 1.01±0.06d | 0.47±0.03g | 29.85 ±0.94efg | 11.88±0.16 | 31.21 ±1.60bc | 11.85±0.28d | 13.86±0.49cd | 0.85±0.02ab |
| S8 | 241.38 ±9.77efh | 76.31 ±1.17af | 1.40±0.11afh | 0.67±0.02e | 28.59 ±1.43cf | 13.95 ±0.07e | 31.64 ±1.06b | 11.73±0.24df | 13.84±0.45cd | 0.85±0.02ab |
| S9 | 266.73 ±5.67ac | 78.31 ±0.82bf | 1.37±0.08ai | 0.71±0.01b | 26.86±0.89c | 14.03 ±0.24e | 29.37 ±0.76de | 11.98±0.32ad | 14.27±0.20abd | 0.84±0.01bdg |
| S10 | 251.46 ±9.27be | 77.63 ±1.35f | 1.45±0.12chi | 0.72±0.03b | 30.79 ±0.75ag | 14.78±0.08 | 30.89 ±0.61be | 11.18±0.45ce | 13.51±0.42c | 0.83±0.05bdf |
| S11 | 205.07 ±9.18d | 58.49 ±0.57e | 1.10±0.03de | 0.44±0.02g | 31.82 ±2.76ag | 11.37±0.03 | 28.56 ±1.25ad | 11.25±0.33ceg | 13.55±0.39c | 0.83±0.02bdf |
| S12 | 228.87 ±1.63hi | 63.94 ±2.06d | 1.20 ±0.05begj | 0.52±0.02c | 36.21 ±0.73dh | 15.40±0.23 | 27.94 ±1.09ad | 11.31±0.11cef | 15.58±0.55 | 0.73±0.03ce |
| S13 | 236.67 ±5.92fi | 67.79 ±3.04ch | 1.31±0.05aj | 0.59±0.02ah | 34.24 ±1.11bh | 12.38 ±0.10d | 28.65 ±1.28ad | 12.91±0.40h | 14.84±0.69b | 0.87±0.02ag |
| S14 | 238.84 ±2.60efi | 68.67 ±1.44gh | 1.19 ±0.12begj | 0.58 ±0.02dfh | 35.21±0.59h | 13.25 ±0.21bc | 28.75 ±0.48ad | 12.59±0.17bh | 14.36±0.19abd | 0.88±0.02ag |

## Analysis the results of OPLS-DA and HCA

Taking the data obtained for VOAV, BAAV, the chemical composition of the volatile oil, the biological characters of the fruits and seed mass of *A. villosum* as variables, the OPLS-DA was used to analyze 14 batches of *A. villosum*. It was found that *A. villosum* in different populations showed a certain degree of separation and aggregation in the model of OPLS-DA. Although S4 and S5 came from Chunwan Town, they were obviously different; even though S13 and S14 were both asexually expanded descendants of *A. villosum* in Chunwan Town, they were separated (Fig 3). This result supported the judgment that the same population of *A. villosum* was composed of different germplasm. After sorting the VIP value of each index of different *A. villosum* fruit and seed mass, it was found that the VIP values of borneol, camphor, α- copalene, (1S, 2E, 6E, 10R)-3, 7, 11, 11-tetramethylbicyclo [8.1.0] undec-2,6-diene, bornyl acetate, fresh weight of 1000 seeds, dry weight of 1000 seeds, transverse and longitudinal were all greater than 1, indicating that they had a significant impact on the classification of different populations, while the impact of other factors were relatively small.

HCA information of *A. villosum* populations analyzed by SIMCA 14.1 were shown in Fig 4, and the fourteen populations of *A. villosum* could be roughly grouped into three categories. The populations S3, S6, S7, and S11 were clustered into one category, S4, S12, S13, and S14 were clustered into another category, and populations S1, S2, S5, S8, S9, and S10 were clustered into the other category. Different populations of *A. villosum* in the same production area were

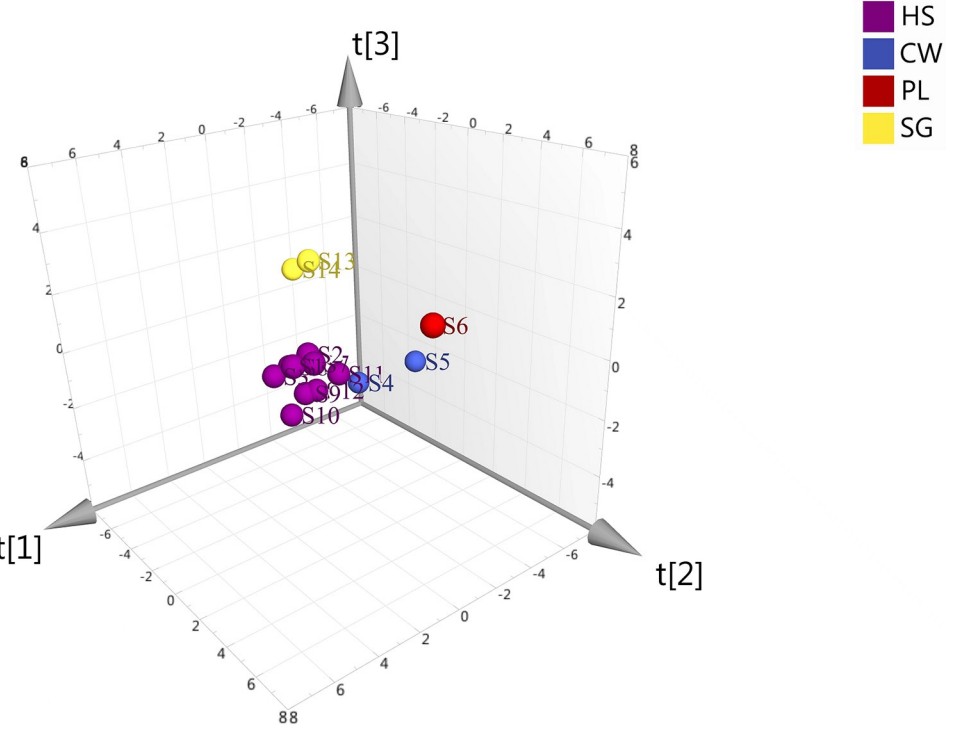

**Fig 3. OPLS-DA map of different populations of *A. villosum*.**

not fully clustered, which might be the result of a combination of ecological and germplasm differences.

## ISSR polymorphism, genetic distance and cluster analysis

The 4 ISSR primers selected were used for PCR amplification of DNA from 14 populations of *A. villosum*, and a total of 23 distinguishable DNA bands were amplified (Fig 5), among which 11 were polymorphic bands, and the proportion of polymorphic bands accounted for 47.8%; the number of polymorphic bands amplified by primer U881 was the most abundant, and the percentage of polymorphic bands accounted for 71.4% (Table 5).

Genetic identity is the main indicator for testing the degree of genetic differentiation and the relationship between groups. The genetic distances of the fourteen populations were between 0.6087 and 0.9565 (Table 6). Among them, the samllest genetic identity was between S14 and S6 populations (0.6087), and the largest were between S9 and S4, S5 and S10, S11 and S13 popula-tions (0.9565). Furthermore, UPGMA clustering map was constructed based on the genetic identity of the 14 populations (Fig 6), and the *A. villosum* populations were divided into four groups at the similarity coefficient of 0.9130. One population, S1, formed a single group. Two populations, S3 and S6, formed one group. Three populations, S2, S7, and S8, formed another group. Eight populations, S4, S5, S9, S10, S11, S12, S13 and S14, formed another group.

## TOPSIS comprehensive quality evaluation

The TOPSIS model was constructed based on the biological properties of the fruit and seed mass of *A. villosum*, VOAV and BAAV, and the relative content of the main components of

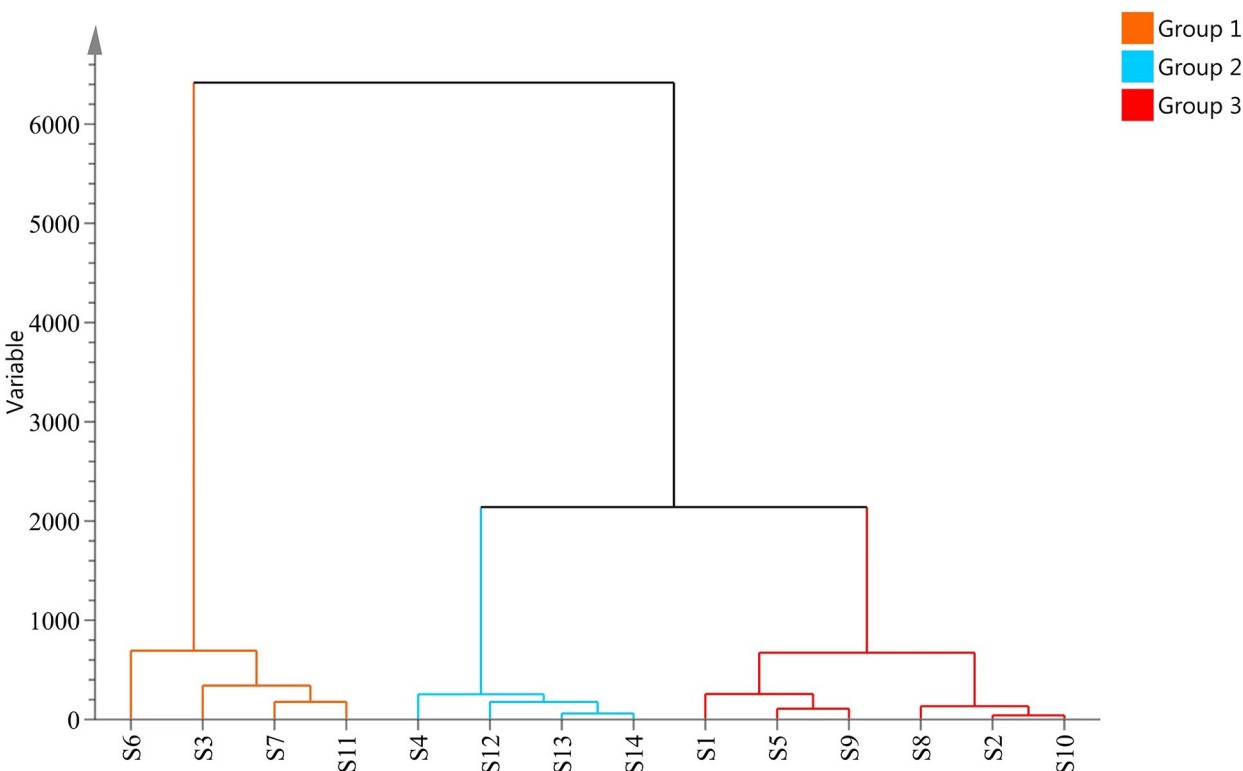

**Fig 4. The cluster analysis of different populations of *A. villosum* based on biological characteristics and main chemical components.**

the volatile oil. The data of each index obtained were processed by formula 1 to perform high-quality index homology processing, and then the entropy value and weight of each index were determined by formula 2, and $E_j$ = (0.3789, 0.9363, 0.8942, 0.9144, 0.9194, 0.9534, 0.9566, 0.9577, 0.9454, 0.8904, 0.9104, 0.9085, 0.8717, 0.9489, 0.9395, 0.9043, 0.9057, 0.9358, 0.8996, 0.8990, 0.9354, 0.8812), $w_j$ = (1.6921, 0.0377, 0.0625, 0.0506, 0.0476, 0.0275, 0.0256, 0.0250,

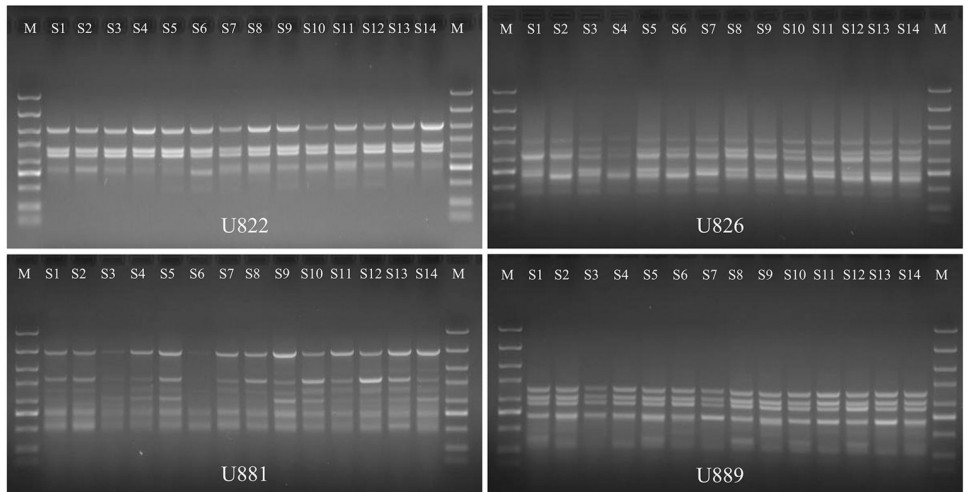

**Fig 5. Amplification results of ISSR-PCR with different primers.**

**Table 5. PCR amplification results by different ISSR primers.**

| Primer | Sequence | Annealing temperature (°C) | Number of amplified bands | Number of polymorphic bands | Polymorphism ratio(%) |
|---|---|---|---|---|---|
| U822 | TCT CTC TCT CTC TCT CA | 50.0 | 4 | 1 | 25.0 |
| U826 | ACA CAC ACA CAC ACA CC | 53.0 | 6 | 3 | 25.0 |
| U881 | GGG TGG GGT GGG GTG | 53.0 | 7 | 5 | 71.4 |
| U889 | DBD ACA CAC ACA CAC AC | 52.0 | 6 | 2 | 33.3 |
| Total | | / | 23 | 11 | 47.8 |

Note: B = (C, G, T); D = (A, G, T).

0.0323, 0.0648, 0.0529, 0.0541, 0.0758, 0.0302, 0.0357, 0.0566, 0.0557, 0.0379, 0.0593, 0.0597, 0.0382, 0.0702).The optimal vector $Z_j^+$(0.0448, 0.0085, 0.0070, 0.0060, 0.0033, 0.0029, 0.0026, 0.0046, 0.0102, 0.0076, 0.0084, 0.0178, 0.0031, 0.0050, 0.0085, 0.0080, 0.0049, 0.0113, 0.0115, 0.0039, 0.0125) and the worst vector $Z_j^-$(0), which were calculated through formula(4) and formula(5).

According to formula (5) and (6), the distance $D$ between the optimal vector $Z^+$ and the worst vector $Z^-$ and the Euclid approach degree $C_i$ with the best solution were calculated, and the fourteen populations was ranked. The result shows that the comprehensive quality ranking of fourteen different populations of *A. villosum* was S5>S2>S4>S9>S1>S10>S12>S6>S13>S8>S14>S3>S11>S7. It can be seen that the two batches of samples (S4, S5) collected in Chunwan Town were of good comprehensive quality, while there were many samples collected in Heshui Town, but of variable quality; while the samples collected from Panlong town (S6) ranked only 8th (moderate), which contradicts the traditional view that Panlong town has the best quality of *A. villosum* (Table 7).

## Discussion

Geoherbalism is one of the criteria for evaluating the quality of traditional Chinese medicine, but geoherbalism is determined by the correspondence among germplasm, environment and origin [32]. In the present study, we have the following findings: First, based on the biological characteristics of fruit and seed mass, the contents of VOAV and BAAV, and the chemical

**Table 6. Genetic identity among *A. villosum* populations.**

| | S1 | S2 | S3 | S4 | S5 | S6 | S7 | S8 | S9 | S10 | S11 | S12 | S13 | S14 |
|---|---|---|---|---|---|---|---|---|---|---|---|---|---|---|
| S1 | 1.0000 | | | | | | | | | | | | | |
| S2 | 0.7826 | 1.0000 | | | | | | | | | | | | |
| S3 | 0.8261 | 0.7826 | 1.0000 | | | | | | | | | | | |
| S4 | 0.6957 | 0.9130 | 0.7826 | 1.0000 | | | | | | | | | | |
| S5 | 0.7826 | 0.8261 | 0.7826 | 0.8261 | 1.0000 | | | | | | | | | |
| S6 | 0.8261 | 0.7826 | 0.9130 | 0.6957 | 0.6957 | 1.0000 | | | | | | | | |
| S7 | 0.8696 | 0.9130 | 0.8696 | 0.8261 | 0.8261 | 0.8696 | 1.0000 | | | | | | | |
| S8 | 0.8696 | 0.8261 | 0.7826 | 0.8261 | 0.8261 | 0.7826 | 0.9130 | 1.0000 | | | | | | |
| S9 | 0.7391 | 0.8696 | 0.8261 | 0.9565 | 0.8696 | 0.7391 | 0.8696 | 0.8696 | 1.0000 | | | | | |
| S10 | 0.8261 | 0.8696 | 0.8261 | 0.8696 | 0.9565 | 0.7391 | 0.8696 | 0.8696 | 0.9130 | 1.0000 | | | | |
| S11 | 0.7826 | 0.9130 | 0.8696 | 0.9130 | 0.9130 | 0.7826 | 0.9130 | 0.8261 | 0.9565 | 0.9565 | 1.0000 | | | |
| S12 | 0.7826 | 0.8261 | 0.7826 | 0.8261 | 0.9130 | 0.6957 | 0.8261 | 0.8261 | 0.8696 | 0.9565 | 0.9130 | 1.0000 | | |
| S13 | 0.7391 | 0.8696 | 0.8261 | 0.8696 | 0.8696 | 0.7391 | 0.8696 | 0.7826 | 0.9130 | 0.9130 | 0.9565 | 0.9565 | 1.0000 | |
| S14 | 0.6957 | 0.7391 | 0.6957 | 0.8261 | 0.9130 | 0.6087 | 0.7391 | 0.8261 | 0.8696 | 0.8696 | 0.8261 | 0.9130 | 0.8696 | 1.0000 |

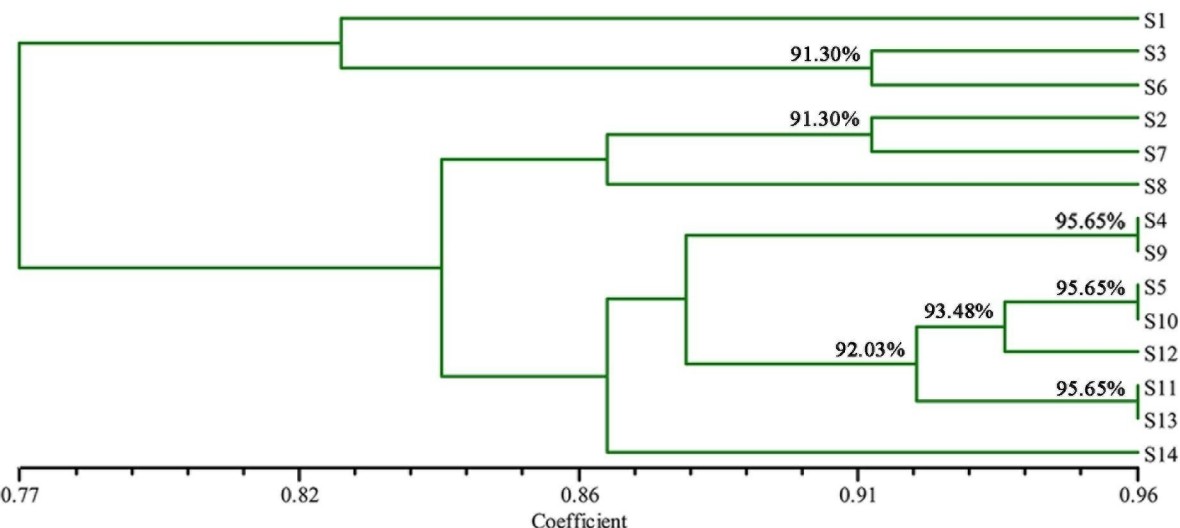

**Fig 6. Genetic similarity of *A. villosum* by UPGMA clustering based on ISSR.**

composition of volatile oils from different populations in the authentic production area, we established a TOPSIS comprehensive quality evaluation model and found the comprehensive quality of *A. villosum* fruits from Chunwan Town (S4, S5) was better, while the comprehensive quality of *A. villosum* fruits from Panlong Town (S6), which is considered to be the best historically, was only moderate among the fourteen samples (S1 and S2 Data). This suggests that there are differences in quality not only between genuine and non-genuine *A. villosum* fruits, but also among fruits from different populations within the genuine producing area.

Many factors influence the genuineness of *A. villosum*, including ecological environment, genetic characteristics, and primary processing [33]. Notably, populations S13 and S14 introduced to Shitang town ranked better than S3, S11 and S7 from Heshui Town in terms of comprehensive quality, indicating that there are still cultivating areas outside the genuine producing areas that can produce good quality fruits of *A. villosum* as long as the germplasm is good, and this finding is a guideline for expanding the cultivating area of *A. villosum* and increasing the total production.

Second, as one of the methods for species identification and genetic diversity analysis, ISSR markers are fast and efficient, high reliability and low cost [34, 35]. The principle of ISSR marker technology lies in the amplification of inter-SSR fragments using anchored primers with 15–24 repetitive bases, and the ISSR technology can provide more genomic information due to the presence of a large number of SSR sequences in the genome. Currently, ISSR

**Table 7. Comprehensive quality evaluation of 14 different populations of *A. villosum*.**

| Number | $D_i^+$ | $D_i^-$ | $C_i$ | Rank | Number | $D_i^+$ | $D_i^-$ | $C_i$ | Rank |
|--------|---------|---------|-------|------|--------|---------|---------|-------|------|
| S1 | 0.0232 | 0.0197 | 0.4591 | 5 | S8 | 0.0257 | 0.0196 | 0.4335 | 10 |
| S2 | 0.0226 | 0.0208 | 0.4792 | 2 | S9 | 0.0237 | 0.0203 | 0.4610 | 4 |
| S3 | 0.0271 | 0.0153 | 0.3609 | 12 | S10 | 0.0247 | 0.0206 | 0.4554 | 6 |
| S4 | 0.0224 | 0.0194 | 0.4651 | 3 | S11 | 0.0294 | 0.0138 | 0.3190 | 13 |
| S5 | 0.0137 | 0.0309 | 0.6934 | 1 | S12 | 0.0233 | 0.0193 | 0.4530 | 7 |
| S6 | 0.0246 | 0.0199 | 0.4470 | 8 | S13 | 0.0229 | 0.0183 | 0.4448 | 9 |
| S7 | 0.0287 | 0.0117 | 0.2903 | 14 | S14 | 0.0250 | 0.0181 | 0.4204 | 11 |

techniques have been widely used in the identification and analysis of Chinese medicines such as *A. tsao-ko* Crevost & Lemariéand [21] and *Alpinia oxyphylla* [36]. Zhang et al. [30] analysed 23 samples of *A. villosum*, *A. longiligulare* T. L. Wu and *A. villosum* Lour. var. *xanthioides* T. L. Wu et Senjen (the variation of *A. villosum*) collected from Guangdong, Yunnan and Hainan Provinces of China using ISSR and found low genetic diversity among the various germplasms. In addition, some studies comparing the volatile constituents of *A. villosum* and *A. villosum*. var. *xanthioides* by GC-MS techniques found that the oil yield was higher in *A. villosum* and the content of bornyl acetate in the volatile oil was much higher than that of *A. villosum*. var. *xanthioides*, and the component with the highest content in the volatile oil of *A. villosum*. var. *xanthioides* is camphor, not bornyl acetate [37, 38]. This may indicate to some extent that *A. villosum* is of better quality than *A. villosum*. var. *xanthioides*, and that differences in the contents of key ingredients can be used as a basis for distinguishing different varieties of Amomi Fructus. In this study, 14 samples of *A. villosum* were analyzed for genetic relationships based on ISRR technique, and it was found that there were significant genetic differences among the samples, and the genetic similarity coefficient of the samples from Panglong (S6) and most of the populations (S2, S4, S5, S8~S14) was less than 0.80. Interestingly, the genetic similarity between S4 and S5, which are from Chunwan Town, is only 0.83; and the genetic similarity between S13 and S14, also from the same plot in Oudong Village, Chunwan Town, is only 0.87 (S3 Data). This indicates that the germplasm in the genuine producing area is diverse and mixed, and two or more different germplasms exist in the same cultivating area. The results of this study show that the germplasm of *A. villosum* is differentiated and rich in diversity in the genuine producing area by being cultivated for thousands of years. We did not focus on the germplasm diversity of *A. villosum* in the non-genuine producing area, such as Yunnan Province, because all *A. villosum* in the non-genuine producing area were introduced from the genuine producing area, and the cultivation history of *A. villosum* in the non-genuine producing area was too short.

Third, the HCA analysis of the quality and biological characteristics of *A. villosum* also found significant differences among fourteen samples. The clustering results of genetic relationships were inconsistent with the clustering results of quality and biological traits, indicating that the quality differences of *A. villosum* in different populations are comprehensively affected by factors such as germplasm and ecological factors. GC-MS can accurately determine the volatile oil content of *A. villosum*. Combining the results of the comprehensive quality and genetic relationship analysis of all samples again, it is revealed that the presence of two or more different germplasms of *A. villosum* are in the same plot in the genuine producing area. Populations S13 and S14 were introduced from the same plot in Chunwan Town of Yangchun City, but their genetic similarity was only 0.8696. Moreover, after they were vegetatively propagated in the same plot in Shitang Town of Shaoguan City, there were significant differences in the chemical composition of volatile oil. S13 could synthesize limonene, while S14 did not contain limonene, indicating that they are two germplasms with significant differences. A similar situation also appeared in populations S4 and S5 from Chunwan Town.

## Conclusion

In this paper, the germplasm diversity of *A. villosum* was studied in order to screen the excellent germplasm of *A. villosum*. The results of this study show that the germplasm diversity of *A. villosum* in the genuine producing area with a long history of cultivation is rich, and the quality differences among various germplasms are obvious, which provides a theoretical basis for the screening of good germplasm in the future.

## Supporting information

**S1 Data. Determination of the relative content of volatile oil components in various populations of *A. villosum* by GC-MS.**
(XLSX)

**S2 Data. TOPSIS method to evaluate the comprehensive quality ranking of 14 populations of *A. villosum*.**
(XLSX)

**S3 Data. Genetic similarity values among the 14 populations.**
(XLSX)

**S1 Raw images.**
(PDF)

## Acknowledgments

We thank Dr. Ruipei Yang (The Fourth Clinical Medical College of Guangzhou University of Chinese Medicine) for his careful reading of the manuscript and his valuable suggestions.

## Author Contributions

**Conceptualization:** Jie Xu, Bohan Yang, Guozhen He.

**Data curation:** Jie Xu, Mingxiao Li.

**Formal analysis:** Yuting Tu.

**Investigation:** Mingxiao Li, Zixiang Li.

**Software:** Zixiang Li.

**Supervision:** Liyun Tang.

**Writing – original draft:** Bohan Yang.

**Writing – review & editing:** Bohan Yang, Guozhen He.

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
