## [Decision Letter · Decision Letter 0]

14 Feb 2022

PONE-D-22-00368Genetic relationship and quality evaluation of Amomum villosum. Lour in genuine producing areaPLOS ONE

Dear Dr. He,

Thank you for submitting your manuscript to PLOS ONE. After careful consideration, we feel that it has merit but does not fully meet PLOS ONE’s publication criteria as it currently stands. Therefore, we invite you to submit a revised version of the manuscript that addresses the points raised during the review process. Please submit your revised manuscript by Mar 31 2022 11:59PM. If you will need more time than this to complete your revisions, please reply to this message or contact the journal office at plosone@plos.org. Please include the following items when submitting your revised manuscript:A rebuttal letter that responds to each point raised by the academic editor and reviewer(s). You should upload this letter as a separate file labeled 'Response to Reviewers'.A marked-up copy of your manuscript that highlights changes made to the original version. You should upload this as a separate file labeled 'Revised Manuscript with Track Changes'.An unmarked version of your revised paper without tracked changes. You should upload this as a separate file labeled 'Manuscript'.

We look forward to receiving your revised manuscript.

Kind regards,

Tzen-Yuh Chiang

Academic Editor

PLOS ONE

Journal Requirements:

Reviewers' comments:

Reviewer's Responses to Questions

**Comments to the Author**

1. Is the manuscript technically sound, and do the data support the conclusions?

Reviewer #1: Partly

Reviewer #2: Partly

2. Has the statistical analysis been performed appropriately and rigorously? 

Reviewer #1: Yes

Reviewer #2: Yes

3. Have the authors made all data underlying the findings in their manuscript fully available?

Reviewer #1: Yes

Reviewer #2: No

4. Is the manuscript presented in an intelligible fashion and written in standard English?

Reviewer #1: Yes

Reviewer #2: Yes

5. Review Comments to the Author

Reviewer #1: The article entitled "Genetic relationship and quality evaluation of Amomum villosum. Lour in genuine producing area" is well written and informative. Some major issues have been raised while reviewing the article:

Why did the authors focus only on bornyl acetates? By the GC-MS study?

Please provide the chromatogram picture for any S1.......... population.

Total number of metabolites:GC MS studies have not been seen in the MS. Authors profile only six compounds. How ___________

Please proved the fruit pictures in results section

How many ISSR primers are used by the authors for ISSR? If there are four, why? (5th table)

The scale is missing in Fig 2.

DPI of fig 3 must be increase.

Add the percentage of similarity to Fig 4.

Discussion part is poorly written in term of GC- MS profiling and ISSR. Kindly elaborate more with other studies.

Kindly write the separate conclusions with translational values.

I have only found molecular data to be novel and interesting. I would like to recommend a major revision of the MS. Authors must provide the details of compounds for GC-MS with peak area, RT, and total number of compounds in a separate table.

Reviewer #2: Overall, this study focused on a topic of having both scientific and practical importance on development of A. villosum, a valuable herbal medicine. A comprehensive analysis on quality assessment as well as genetic diversity was performed. However, a strong concern about the present study might locate on its small germplasm size, which couldn’t well support a solid conclusion. Detailed comments are listed below.

Major conmments

1. The current title of the MS didn’t well fit its contents, since most of this study focused on quality evaluation rather than genetic relationship.

2. At the end of the Abstract, “Our data suggest that the germplasm of A. villosum is abundant in genuine producing area and that it is possible to select good germplasm resources in the area, expand the planting areas in non-genuine producing area and use a single germplasm resource for cultivation to obtain more quality products”, I can’t agree with that the results showed in Abstract could solidly and directly support its conclusion. I would suggest the author re-write the abstract, emphasizing on its highlight results clearly. Additionally, a brief background summary (e.g. introduction of genuine producing area) of the species is necessary in the beginning of Abstract section. Without background summary, readers might be confused about the results demonstrated in the Abstract.

3. Not only cultivation background but also research progresses of chemical components and genetic diversity of A. villosum should be briefly but clearly introduced in Introduction section.

4. Sampling information was incomplete. The author should add details of the sampling cite (latitude and longitude) when studying on genetic diversity analysis.

5. The authors should clarify how many individuals they collected for each population. E.g. does S1 means one individual or several mixed samples from the same population? If yes, in areas 2,3&4, only one or two populations were collected for each area, resulting in a small germplasm size. Usually, a larger sample size would be acceptable in a genetic diversity study.

6. “The fresh weight, dry weight, seed mass, seed mass dry weight, thousand-grain fresh weight (seed), and thousand-grain dry weight (seed) of each population of A. villosum dried fruit were measured by electronic balance”, here is a confusing expression, how did the author measure the fresh weight using dried fruits?

7. ISSR analysis in this MS seemed to be relatively independent. Results in this part didn’t show highlight significance to the whole study. Deeper discussions on genetic relationship with their quality evaluation would definitely make the study more valuable and interesting.

8. “The TOPSIS model was constructed based on the biological properties of the fruit and seed mass of A. villosum, VOAV and BAAV, and the relative content of the main components of the volatile oil”, based on what exact kind of biological properties did the authors determine as “good quality indicators? E.g. large seed mass size or high level of active component? The authors should explain in detail on the evaluation model parameters.

9. “Third, after combining the comprehensive quality and genetic relationship analysis results of all samples, it is found that there are two or more different species of A. villosum in the same plot in the genuine producing area”, could the authors give the conclusion that S13&14 or S4&5 are different “species” based on evidences from the current study?

Minor comments

1. Line and page numbers should be added to the MS for reviewers to give comments.

2. Full names should be presented when they first occurred in the MS. Revisions are needed in the Abstract and introduction sections, including but not limited to “A. villosum”, “ISSR”, “UPGMA”, “TOPSIS model”, and “RAPD”.

3. Minor changes are needed to suit grammar for the several sentences:

• The pharmacodynamic material basis of A. villosum is volatile oil, which mainly contains camphene, camphor, bornyl acetate and other chemical components, of which bornyl acetate is the most significant active component, which has pharmacological effects such as sedative and analgesic, anti-inflammatory and anti diarrhea, inhibition of fatty liver” and “but whether the diversity of germplasm is still abundant or not remains to be studied.

• The relative content of the main components of the volatile oil in each A. villosum population, the biological characteristics of the fruit and seed clusters, VOAV and BAAV were used as variables, combining with the orthogonal partial least squares discrimination analysis (OPLS-DA) and hierarchical clustering analysis (HCA) in the SIMCA 14.1 software to analyze the quality differences among 14 samples of A. villosum (Figure 1 & Figure 2).

• The relative content of the main components of the volatile oil in each A. villosum population, the biological characteristics of the fruit and seed clusters, VOAV and BAAV were used as variables, combining with the orthogonal partial least squares discrimination analysis (OPLS-DA) and hierarchical clustering analysis (HCA) in the SIMCA 14.1 software to analyze the quality differences among 14 samples of A. villosum (Figure 1 & Figure 2).

4. The authors should go though the whole MS for minor writing errors. E.g. in Introduction, “qi” should be “Qi”; In Material and methods section, “Traditional” should be “traditional”; A “,” is needed before “respectively”, etc.

6. PLOS authors have the option to publish the peer review history of their article (what does this mean?). If published, this will include your full peer review and any attached files.

Reviewer #1: No

Reviewer #2: No

---

## [Author Response · Author response to Decision Letter 0]

29 Mar 2022

Thank you for your valuable and thoughtful comments and these comments are all valuable and very helpful for revising and improving our paper, as well as the important guiding significance to our researches. We have incorporated them into our paper. All your questions have been answered point by point. Thank you very much again for your efforts on our manuscript. If you have any questions, please let us know and we are willing to make further revisions if necessary.

---

## [Decision Letter · Decision Letter 1]

18 Apr 2022

PONE-D-22-00368R1Research on germplasm diversity of Amomum villosum. Lour in genuine producing areaPLOS ONE

Dear Dr. He,

Thank you for submitting your manuscript to PLOS ONE. After careful consideration, we feel that it has merit but does not fully meet PLOS ONE’s publication criteria as it currently stands. Therefore, we invite you to submit a revised version of the manuscript that addresses the points raised during the review process.

We look forward to receiving your revised manuscript.

Kind regards,

Tzen-Yuh Chiang

Academic Editor

PLOS ONE

Journal Requirements:

Reviewers' comments:

Reviewer's Responses to Questions

**Comments to the Author**

1. If the authors have adequately addressed your comments raised in a previous round of review and you feel that this manuscript is now acceptable for publication, you may indicate that here to bypass the “Comments to the Author” section, enter your conflict of interest statement in the “Confidential to Editor” section, and submit your "Accept" recommendation.

Reviewer #1: All comments have been addressed

Reviewer #2: All comments have been addressed

2. Is the manuscript technically sound, and do the data support the conclusions?

Reviewer #1: Yes

Reviewer #2: Yes

3. Has the statistical analysis been performed appropriately and rigorously? 

Reviewer #1: Yes

Reviewer #2: Yes

4. Have the authors made all data underlying the findings in their manuscript fully available?

Reviewer #1: Yes

Reviewer #2: Yes

5. Is the manuscript presented in an intelligible fashion and written in standard English?

Reviewer #1: Yes

Reviewer #2: Yes

6. Review Comments to the Author

Reviewer #1: Dear Authors please add some important points in the revised MS.

Minor comments

previous Q.

How many ISSR primers are used by the authors for ISSR? If there are four, why? (5th table)

Response: Thanks for the constructive comment. A total of 10 ISSR primers were selected for this study according to previous studies that used ISSR techniques to identify germplasm of A. villosum. Finally, we selected four primers with clear bands, obvious spacing and good polymorphism for this experiment [8][10].

New suggestions: Response in MS: Twenty-three selected ISSR primers were used in the germplasm identification study of A. villosum2930, line number 242-243, Please re-check and correct in the MS.

Q. Discussion part is poorly written in term of GC- MS profiling and ISSR. Kindly elaborate more with other studies.

Response: Thank you for the valuable suggestions. The discussion section has been reformulated with other GS-MS and ISSR-related studies as suggested, as detailed in the revised manuscript.

New suggestions-Authors improved some part of this section, but I will suggest kindly elaborate more with previous study.

Conclusions: Please add some points related to chemotypes and its variations (GC-MS study).

Reviewer #2: The MS has been much improved in the current vision, and most of the comments have been addressed. Several minor comments are listed below.

1. Line 14, “consistent quality” should be “consistently qualified”.

2. Line 21-24, move “In this study, we selected 12 populations… for quality evaluation and genetic variation detection” to “methods” section.

3. Line 339&345, pay attention to the reference number format errors.

7. PLOS authors have the option to publish the peer review history of their article (what does this mean?). If published, this will include your full peer review and any attached files.

Reviewer #1: No

Reviewer #2: No

---

## [Author Response · Author response to Decision Letter 1]

20 Apr 2022

Thank you very much for your valuable and thoughtful comments. In response to your suggestion, we have added relevant content to the revised manuscript and made revisions to parts of it.Those comments are all valuable and very helpful for revising and improving our paper, as well as the important guiding significance to our researches. Thank you very much again for your efforts on our manuscript.

---

## [Decision Letter · Decision Letter 2]

26 Apr 2022

Research on germplasm diversity of Amomum villosum. Lour in genuine producing area

PONE-D-22-00368R2

Dear Dr. He,

We’re pleased to inform you that your manuscript has been judged scientifically suitable for publication and will be formally accepted for publication once it meets all outstanding technical requirements.

Kind regards,

Tzen-Yuh Chiang

Academic Editor

PLOS ONE

Additional Editor Comments (optional):

Reviewers' comments:

Reviewer's Responses to Questions

**Comments to the Author**

1. If the authors have adequately addressed your comments raised in a previous round of review and you feel that this manuscript is now acceptable for publication, you may indicate that here to bypass the “Comments to the Author” section, enter your conflict of interest statement in the “Confidential to Editor” section, and submit your "Accept" recommendation.

Reviewer #1: All comments have been addressed

Reviewer #2: All comments have been addressed

2. Is the manuscript technically sound, and do the data support the conclusions?

Reviewer #1: Yes

Reviewer #2: Yes

3. Has the statistical analysis been performed appropriately and rigorously? 

Reviewer #1: Yes

Reviewer #2: Yes

4. Have the authors made all data underlying the findings in their manuscript fully available?

Reviewer #1: Yes

Reviewer #2: Yes

5. Is the manuscript presented in an intelligible fashion and written in standard English?

Reviewer #1: Yes

Reviewer #2: Yes

6. Review Comments to the Author

Reviewer #1: Dear Author

Please add some points related to chemotypes and its variations (GC-MS study). Still not well written in the revised MS. I agree with with the justifications but authors may write about the translational value of the the MS.

Reviewer #2: In the current revision, all my comments have been addressed. I have no more questions and agree with its publication on PlosOne.

7. PLOS authors have the option to publish the peer review history of their article (what does this mean?). If published, this will include your full peer review and any attached files.

Reviewer #1: No

Reviewer #2: No

---

## [Editor Report · Acceptance letter]

22 Aug 2022

PONE-D-22-00368R2 

Research on germplasm diversity of *Amomum villosum*. Lour in genuine producing area 

Dear Dr. He:

I'm pleased to inform you that your manuscript has been deemed suitable for publication in PLOS ONE. Congratulations! Your manuscript is now with our production department. 

Kind regards, 

on behalf of

Dr. Tzen-Yuh Chiang 

Academic Editor

PLOS ONE